# Roles of Different Signaling Pathways in *Cryptococcus neoformans* Virulence

**DOI:** 10.3390/jof10110786

**Published:** 2024-11-13

**Authors:** Fawad Mahmood, Jun-Ming Chen, Ammar Mutahar Al-Huthaifi, Abdullah Ali Al-Alawi, Tong-Bao Liu

**Affiliations:** 1Medical Research Institute, Southwest University, Chongqing 400715, China; fawadsawabi123@gmail.com (F.M.); ammaralhuthaifi@gmail.com (A.M.A.-H.); alalawisky@tu.edu.ye (A.A.A.-A.); 2State Key Laboratory of Resource Insects, Southwest University, Chongqing 400715, China; cjm0712@email.swu.edu.cn; 3Jinfeng Laboratory, Chongqing 401329, China; 4Engineering Research Center for Cancer Biomedical and Translational Medicine, Southwest University, Chongqing 400715, China

**Keywords:** *Cryptococcus neoformans*, signaling pathways, morphological development, pathogenicity

## Abstract

*Cryptococcus neoformans* is a widespread fungal pathogen that can infect the human central nervous system (CNS) and cause fungal meningitis, leading to hundreds of thousands of deaths worldwide each year. Previous studies have demonstrated that many signal transduction pathways are crucial for the morphological development and virulence of *C. neoformans*. In this review, data from over 116 research articles have been compiled to show that many signaling pathways control various characteristics of *C. neoformans*, individually or in association with other pathways, and to establish strong links among them to better understand *C. neoformans* pathogenesis. Every characteristic of *C. neoformans* is closely linked to these signaling pathways, making this a rich area for further research. It is essential to thoroughly explore these pathways to address questions that remain and apply a molecular mechanistic approach to link them. Targeting these pathways is crucial for understanding the exact mechanism of infection pathogenesis and will facilitate the development of antifungal drugs as well as the diagnosis and prevention of cryptococcosis.

## 1. Introduction

*Cryptococcus neoformans* is a ubiquitous pathogen found worldwide, primarily in pigeon guano and swamps [1,2,3]. It can infect a wide range of organisms and is acquired through inhalation of spores, which initiates lung disease [4,5,6]. The emergence of fungal pathogens poses a significant threat to global public health [7,8]. The spread of cryptococcosis begins with a lung infection and then progresses to the bloodstream, allowing it to reach the brain and CNS [9]. In immunocompetent hosts, *C. neoformans* infections typically result in mild or asymptomatic presentation and are efficiently cleared by the immune system [10]. Severe disease manifests in the absence of cell-mediated immunity, as observed with AIDS or organ transplant recipients on immunosuppressive therapy. In these immunocompromised patients, *C. neoformans* commonly causes meningoencephalitis, a life-threatening central nervous system infection if left untreated [11]. *C. neoformans* expresses various phenotypes, including virulence traits such as the capsule, melanin production, and the ability to thrive at 37 °C. The polysaccharide capsule is a key virulence factor of *C. neoformans*, accounting for about 25% of its overall virulence potential. The capsular structure is induced under specific host-associated conditions, including neutral or alkaline pH, elevated carbon dioxide concentrations, and iron limitation. The capsule enhances virulence through numerous mechanisms that protect *C. neoformans* from host immune defenses. It is a potent antiphagocytic barrier, inhibiting the uptake of yeast cells by phagocytes in vitro unless opsonized by antibodies or complement. Additionally, the capsule interferes with host immune processes, promoting the fungus’ persistence and survival. Melanin is a key virulence factor for *C. neoformans*, providing significant protection against environmental stresses such as ultraviolet radiation and reactive oxygen species. It facilitates the fungus’ survival within macrophages by effectively neutralizing free radicals. Additionally, melanin confers resistance to antifungal drugs [12]. Effective communication with the environment is crucial for all living organisms, and fungi utilize intricate signaling systems to regulate their proliferation, development, and, in some cases, virulence.

The signaling pathways play a key role in controlling the expression of these phenotypes and other essential cellular processes [1,13,14], particularly in the context of pathogenic microorganisms, which must adapt from the environment to the host environment and mount an appropriate response to establish an infection. Host conditions present challenges such as variations in nutrient availability, oxygen levels, pH, temperature, and threats posed by the immune response. Signal transduction pathways are critical for mediating microbial adaptation [15]. Numerous pathogenic fungi in our world cause either life-threatening diseases in humans or significant economic damage to crop production. The ability to sense host conditions is crucial for fungi to proliferate during infection, and signal transduction pathways play a critical role in sensing environmental conditions and facilitating adaptation.

Signal transduction pathways have been found to regulate fungal morphogenesis, and some of these pathways are associated with the virulence of pathogenic fungi. These pathways can control cell shape and react to changes in the environment. Our current focus is on understanding the factors that contribute to the evolution of *Cryptococcus* pathogenic species. The current study reviews our knowledge of the complex connections that lead to human disease. Specifically, our review focuses on various signaling pathways, including protein kinase C (PKC), mitogen-activated protein kinase (MAPK), the cell wall integrity pathway, protein kinase A/cyclic AMP (cAMP), RAS1, and RAS2 signal transduction pathways, and the calcium–calcineurin signaling pathway.

## 2. Protein Kinase A/cAMP Signaling Pathway

Cellular organisms respond to environmental cues through complex signal transduction pathways. In higher eukaryotic organisms, environmental stimuli are detected by transmembrane receptors such as G-protein-coupled receptors (GPCRs). These receptors activate adenylate cyclase (AC) through RAS proteins or G proteins, leading to the production of a second messenger called cyclic adenosine monophosphate (cAMP). When cAMP binds to the protein kinase A (PKA) tetramer regulatory subunits (PKR), they separate from the catalytic subunits, causing the catalytic subunits to become active and regulate downstream signaling and gene transcription [16,17,18,19]. The cAMP-PKA pathway transmits environmental signals and enables various cellular functions [20]. Despite diverse regulation in fungi, the components involved in the cAMP-PKA pathway are highly conserved across different fungi. Fungal cAMP signaling pathways are quite versatile despite the differential regulation of similar processes [21]. Most eukaryotic cell types have a critical regulatory role for cyclic AMP. However, there can be significant variations in the cAMP-regulated pathways between different organisms and even within tissues of a single organism [22] (Table 1).

Cellular cAMP levels depend on the activities of degradative enzymes, phosphodiesterases, and the biosynthetic enzyme adenylyl cyclase. Heterotrimeric G proteins regulate the adenylyl cyclase enzyme [21]. The cAMP/PKA pathway is crucial for the virulence of *C. neoformans* in animal hosts and has received significant attention. In *C. neoformans*, three main factors include a polysaccharide capsule, deposition of melanin in the cell wall, and the ability to multiply at 37 °C [37]. The genes responsible for *C. neoformans*’ melanin and capsule are co-regulated at the transcription level through the cyclic AMP pathway. In cases where *C. neoformans* strains have defects in the cAMP signaling cascade, they are unable to produce capsule or melanin, resulting in significantly reduced virulence [38].

The G-protein signaling pathway in *C. neoformans* consists of three subunits of the G protein (Gpa1, Gpa2, and Gpa3) and one subunit of the G protein (Gpb1) [39]. Gpa1, along with the adenylyl cyclase-associated protein Aca1, regulates the cAMP-protein kinase A (PKA) signaling and plays a crucial role in producing two important virulence factors, capsule and melanin [21,40]. Additionally, Gpa1 is also involved in the mating process of *C. neoformans*. On the other hand, the roles of G-protein subunits Gpa2 and Gpa3 are less well defined. In a different context, the sole G subunit Gpb1 is vital for pheromone sensing during mating through the Cpk1 MAP kinase pathway in *S. cerevisiae* [41].

Among transmembrane receptors, the largest family consists of G-protein-coupled receptors (GPCRs), which are crucial in transducing extracellular signals into intracellular responses and relaying signals. GPCRs are sensors that respond to diverse stimuli such as protons, light, calcium, amino acids, odorants, nucleotides, polypeptides, steroids, proteins, and fatty acids [42]. They are essential for regulating intracellular responses and controlling cell function through both G-protein-dependent and independent pathways [43]. Despite their significance in signaling regulation, few GPCRs apart from pheromone receptors have been thoroughly studied in most fungal systems [44]. Pheromone receptors like Ste3/Cpr and Ste3a/Cpra, expressed by α mating type cells, have been identified as crucial for mating, with Cpra possibly impacting *C. neoformans* virulence [45]. A pheromone receptor-like GPCR (Cpr2) has also been found in *C. neoformans*. However, its gene is not mating type-specific, and its functions remain unknown [46]. Remarkably, the cAMP/PKA pathway regulates melanin formation, capsule size, virulence, and other traits such as mating [21,47]. The pathway components are categorized as shown in Figure 1.

The cAMP/PKA pathway plays a critical role in the pathogenesis of *C. neoformans*. Mutants with defects in pathway constituents such as gpa1, cac1, and pka1 exhibited altered virulence in a mouse model of cryptococcosis. Specifically, these mutants showed reduced capsule formation, melanin production, fertility, and virulence [39]. Upstream components of the pathway, including a candidate G-protein coupled receptor (Gpr4), a Gα protein (Gpa1), a G beta-like/RACK1 homolog (Gib2), and a regulator of G-protein signaling (RGS) protein (Crg2) [39,40], as well as carbonic anhydrase Can20, play crucial roles in controlling adenylyl cyclase (Cac1) and cAMP production in response to carbon dioxide, amino acids, and glucose [37]. Understanding the downstream effects of the cAMP/PKA pathway on specific phenotypes has been facilitated by transcriptional profiling of mutants with defects in pathway components. For example, a microarray study comparing the transcriptomes of a wild-type strain with a *gpa1*Δ mutant was conducted [48,49].

**Figure 1 jof-10-00786-f001:**
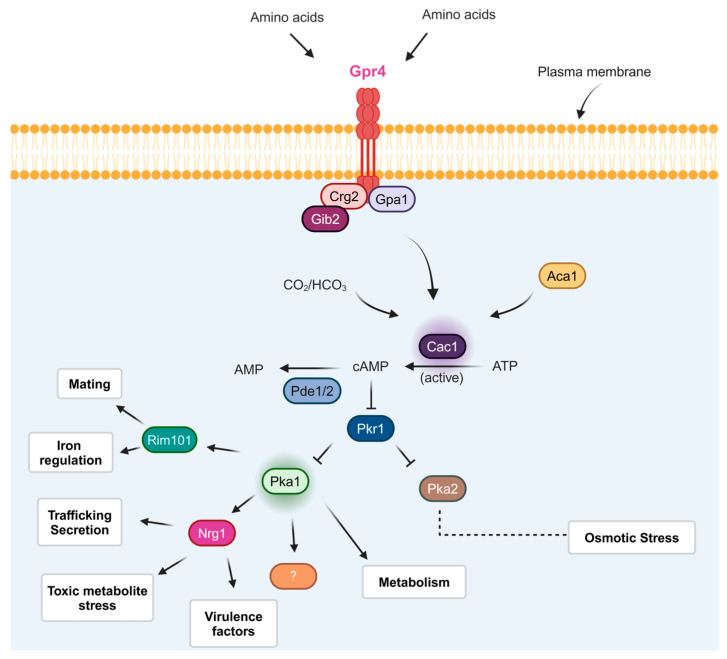
A model of the elements and downstream targets of the cAMP/protein kinase A (PKA) pathway in *C. neoformans*. The components upstream of the intended pathway are associated with the cell membrane and respond to external signals such as glucose and amino acids [39,49]. The components include the G-protein-coupled receptor Gpr4, a regulator of G-protein signaling Crg2, a Gbeta-like/RACK1 homolog Gib2, and the G-protein alpha subunit Gpa1 [46,49]. Together, these proteins drive the activity of adenylyl cyclase (Cac1) and the Aca1 protein, as well as CO_2_/HCO_3_^−^, ultimately affecting cAMP levels [40]. Additionally, phosphodiesterases (Pde1/2) can reduce cAMP levels, and signals can also modulate the function of protein kinase A (PKA) by affecting the levels of cAMP, which binds the regulatory subunit (Pkr1) and causes it to dissociate from the catalytic subunits Pka1 and Pka2 [21,50]. Transcription factors Rim101 and Nrg1 regulate some of these processes, although not all direct connections are recognized in every case [51]. This pathway is illustrated by arrows from Pka1, with the potential involvement of other regulatory factors suggested by the protein marked with a question mark.

Furthermore, deletion of the gene encoding phosphoglucose isomerase Pgi1 in *C. neoformans* results in the disruption of melanin and capsule formation. However, these traits can be restored by adding external cAMP [52]. On the other hand, the genes encoding trehalose synthesis, *TPS1*, and *TPS2*, are crucial for the development of virulence factors such as capsule, melanin, thermotolerance, control of protein secretion, mating, and cell wall integrity in *C. gattii* [53]. Tsp2 has also been suggested to act as a possible glucose sensor, partly based on the finding that Tsp2 negatively regulates melanin formation, a phenotype that is suppressed by glucose [14]. Regulation of the capsule involves both the cAMP/PKA pathway and the iron regulator Cir1, a zinc finger transcription factor [21]. A study has revealed that this pathway controls the transcript levels for some genes required for capsule production, as well as two genes, *LAC1* and *LAC2*, responsible for encoding the laccases that produce melanin. The study also suggests that the gene coding for the expected transcription factor, Nrg1, is controlled via Gpa1 and the cAMP/PKA pathway. A subsequent comprehensive study of Nrg1 has shown that the protein, containing a C_2_H_2_ zinc finger domain, plays a role in capsule production, virulence, and mating [50].

It is likely that the cAMP/PKA pathway affects the expression of various virulence functions, such as the capsule, through multiple downstream mechanisms. In a transcriptional analysis of *gpa1* and *nrg1* mutants using sequential analysis of gene expression (SAGE), capsule-linked genes associated with the secretory pathway were discovered [54]. However, it is not yet clear how PKA might impact the expression of components in the secretory pathway. Regulation may influence capsule production at several levels, including gene expression, protein synthesis, RNA degradation, and protein localization. Additionally, the transcriptional effect of PKA mutations may be the result of an indirect influence on the secretory pathway, for example, through the unfolded protein response [55].

In *C. neoformans*, the pathway linking nutrient sensing to downstream functions and signaling pathways has been altered to adapt the cell surface and express different virulence elements such as melanin and capsule. Clues suggest that glycolysis is linked to the cAMP/PKA signaling pathway in *C. neoformans*, but direct links have not been established. Mutants with reduced glycolysis due to the absence of hexokinase genes have shown decreased virulence in cryptococcosis infection in animal models [15]. Additionally, removing the phosphoglucose isomerase gene *PGI1* from *C. neoformans* halted the production of important virulence factors such as capsule and melanin. These effects were reversed after the addition of exogenous cAMP. Furthermore, *TPS1* and *TPS2* genes responsible for trehalose formation are required for the amplification of virulence factors, thermotolerance, protein secretion control, cell wall integrity, and mating in *Cryptococcus* species such as *C. gattii* [53]. Taken together, these findings suggest the involvement of glycolysis in cAMP/PKA pathway activation, although further research focused on the function of fructose 1,6-bisphosphate is needed.

The involvement of this pathway in *C. neoformans*’ ability to cause cryptococcosis is evidenced by studies showing that various mutants with defects in key components like Gpa1, Pka1, Aca1, and Cac1 exhibit reduced virulence in mouse models. The cAMP signaling cascade in *C. neoformans* influences its virulence by regulating important virulence factors, including the capsule, enzymes such as laccase, proteases, superoxide dismutase, and urease, as well as essential metabolic functions for growth. The cAMP signaling cascade also plays a crucial role in mating and virulence in *C. neoformans* (Figure 2). Mutant strains lacking the serotype A protein kinase A catalytic subunit Pka1 are incapable of mating, fail to produce capsule or melanin, and are nonvirulent in animal models. Conversely, mutant strains lacking the regulatory subunit PKA, Pkr1, display hypervirulence and excessive capsule production [56]. Pathogenic fungi can undergo morphological changes and alter their cell size during infection, providing them with a significant advantage. Similarly, *C. neoformans* undergoes morphological changes during infection, such as polyploidization and alterations in capsule size, which enhance the fungus’s resistance to phagocytosis and various types of stress. The cAMP/PKA signaling pathway regulates morphological changes. Recent articles have provided detailed insights into titan cell development and have established in vitro methods for generating these cells [57]. Ras1 and Ras2 proteins have been examined in the fungus and shown to have various shared roles in mating filamentation and mating under thermal stress [58]. Notably, a two-hybrid yeast assay demonstrated that Ras1 interacts with Cac1 over its putative RA domain and with Gib2, suggesting that Gib2 may regulate the cAMP signaling pathway protein Ras1 and Cac1 [59]. The endosomal sorting complex required for transport (ESCRT) apparatus is involved in heme trafficking in *C. neoformans*, and functional studies have confirmed its roles in capsule development and iron acquisition. In a *pkr1* mutant, a deficiency in the ESCRT protein Vps23 led to a decrease in the otherwise enlarged capsule, as the ESCRT acted downstream of the cAMP/PKA signaling pathway [54]. These results suggest a strong connection between the cAMP/PKA pathway, capsule formation, and iron sensing in *C. neoformans*. Even though the exact mechanisms connecting iron sensing to the cAMP/PKA pathway are unclear, changes in intracellular iron levels or the labile iron pool probably trigger various signaling pathways, including cAMP/PKA, to promote survival and adaptation [15].

Another important role of the cAMP/PKA pathway in *C. neoformans* is promoting the formation of titan cells, a morphological change that significantly enhances the fungus’ resistance to phagocytosis [60]. The exact molecular mechanisms by which iron sensing connects to the cAMP/PKA pathway in *C. neoformans* remain unclear. Expounding these molecular mechanisms could find novel regulatory elements in the signaling pathway, possibly revealing new targets for therapeutic purposes. Further research in this area is important to fully understand how iron sensing affects the signaling processes that determine virulence in this fungus. In the context of PKA target identification, although quite a lot of targets of PKA phosphorylation have been documented, a comprehensive overview of all direct targets is still incomplete. Defining these targets is crucial for elucidating the molecular mechanisms by which the PKA pathway controls different virulence features. Further studies are essential to fully understand how the cAMP/PKA pathway interacts with other signaling networks, particularly those related to stress responses and nutrient sensing. This understanding is key to unraveling how these signaling pathways synergistically affect virulence.

The downstream targets of PKA are critically important. Therefore, it is crucial to understand the links between environmental sensing and the ability of this fungus to survive and proliferate in hosts. Transcriptional profiling has been beneficial in revealing connections with transcription factors and other proteins that play a role in regulating virulence factor expression, the secretory pathway, and the iron and pH regulatory networks of the stress response. Phenotypic studies have also identified additional regulators of capsule formation, such as the Ova1 protein, and established a connection between cAMP/PKA signaling and the production of enlarged cells [37].

The existing research gaps concerning the cAMP/PKA signaling pathway highlight several critical areas requiring further investigation, including the limited understanding of the molecular mechanisms of this pathway, the necessity for the identification of specific phosphorylation targets, and the exploration of its interactions with other signaling pathways as well as its roles in glycolysis and iron sensing. Furthermore, to build on successful approaches to date, it is evident that new tools and systems biology approaches are needed to gain a detailed molecular understanding of cAMP/PKA signaling.

## 3. Cell Wall Integrity Pathway (CWI)

The cell wall of *C. neoformans* is a vital structure primarily made up of chitin, glucans, chitosan, and glycoproteins. These components help maintain the integrity and rigidity of the cell wall [61,62,63], which plays a key role in protecting the cell from various types of stress [64]. Furthermore, the cell wall facilitates interactions with the external environment through various receptors. Activation of these receptors triggers different signaling cascades within the fungal cell [65].

In the 1990s, a study on the yeast *S. cerevisiae* showed that the CWI pathway acted as the primary MAPK cascade. This pathway is crucial for assessing the structural and functional condition of the yeast cell envelope and for initiating a suitable response when its integrity is compromised [66]. The CWI pathway is responsible for forming and regulating the cell wall in fungi, and it was initially thought to be only activated in response to “cell wall problems”. However, it also influences the expression and production of certain important molecules that help fungi compete more effectively in their environment [65,67]. This pathway has been extensively studied in *S. cerevisiae* and is well conserved among fungi [68]. Although the cell wall structure and CWI are similar in all fungi, many aspects of this pathway are still unknown in the case of the *C. neoformans* fungus [66,69]. Furthermore, this pathway is very similar to the PKC pathway in organisms. The cellular integrity pathway is important for aging, oxidative stress responses, and cellular morphogenesis. It is regulated by different receptors and factors, which ultimately initiate the MAPK (mitogen-activated protein kinase) module (Figure 3) [66].

Cell wall integrity is essential for vital cellular processes, including fungal growth, survival, and pathogenesis. The MAPK signaling cascade plays a crucial role in adaptation to environmental stresses [70]. The polysaccharide capsule is the most prominent characteristic of *C. neoformans*. It primarily comprises two types of polysaccharides: glucuronoxymannan (GXM), which makes up 90–95%, and glucuronoxylomanogalactan (GXMGal), which constitutes 5–8%. Additionally, small amounts of mannoproteins are also present [71]. Although the capsule is a fundamental structure of *C. neoformans*, the mechanisms involved in its production are still not completely understood. The prevailing model suggests that the core components are produced intracellularly in the endoplasmic reticulum and then transported to the extracellular space in vesicles. Subsequently, the vesicle contents are released, and the polysaccharide fibers adhere to the cell wall, particularly the α-1,3-glucan. Currently, many genes and several transduction pathways, such as the cAMP-PKA pathway and MAPK pathways like Hog1, are known to be involved in capsule synthesis. New studies also suggest that the MAPK Bck1 module may play a role in regulating capsule production [72].

The cell wall is crucial for viability and pathogenesis, and the CWI signaling pathway primarily regulates its biosynthesis and repair. A study found that deleting any of the three core kinases, BCK1, MKK2, or MPK1, in the CWI signaling pathway affected not only the cell wall but also the number of capsules. Deletion of any of the kinases led to significantly reduced cellular cAMP levels, and adding exogenous cAMP levels rescued the capsule defect and some cell wall defects, confirming that the cAMP/protein kinase pathway and the CWI system work together to regulate the capsule [72,73,74]. The upstream structure of cryptococcal CWI signaling is still unknown [75]. Various stressors can activate the CWI signaling pathway, regulating multiple transcription factors and triggering an adaptive response. Many genes involved in cell wall formation are induced through the transcription factors ScRlm1 and the ScSwi4/ScSwi6 complex, as observed in classical studies on *S. cerevisiae* transcription factors’ activation via the CWI pathway [76]. Additionally, CNAG_01438 may be involved in the response and activation of the CWI pathway [75].

A *grx4*Δ mutant showed increased sensitivity to SDS, caffeine, and calcofluor white, indicating that Grx4 is crucial in maintaining membrane CWI and responding to caffeine [77]. A study reported that the CWI pathway does not affect changes in the transcriptome, but it might be involved in the posttranslational modification of various transcription factors responsible for masking (Figure 4) [78].

Diphenolic compounds, such as L-DOPA, rely on the diphenol oxidase enzyme, which is encoded by two genes. *LAC1* and *LAC2* regulate the synthesis of melanin production, with the *LAC1* gene being the primary melanin producer [48]. These enzymes confine the melanin that accumulates in the cell wall, contributing to the survival of *C. neoformans* and preserving the integrity of its structure. Moreover, melanin is important for virulence and dissemination from the lungs to other areas.

The downstream mechanisms of the CWI signaling pathway in *C. neoformans* are well known, but the upstream signals that initiate the CWI pathway are still largely unknown. Identifying the molecular targets and initial initiation triggers of the CWI pathway is important for the understanding of how *C. neoformans* responds to environmental stressors. Post-translational modifications (PTMs) such as phosphorylation, ubiquitination, and simulation play key roles in modulating the stability, activity, and subcellular localization of CWI pathway proteins, which can significantly affect the virulence of *C. neoformans* [47,79,80]. Despite advances in understanding the CWI in *C. neoformans*, many direct targets of this pathway still need to be discovered. More research is needed to identify these targets and clarify their contributions to fungal virulence. Additionally, further research is needed to understand how the CWI pathway interacts with other signaling pathways in *C. neoformans*. Investigating these interactions will improve our understanding of the complex molecular regulatory networks that support fungal pathogenesis. Furthermore, studying the role of the CWI pathway in antifungal resistance mechanisms within *C. neoformans* may lead to the discovery of novel therapeutic targets and strategies. Therefore, the CWI pathway is essential for the intrinsic adaptation of antifungals and is crucial for factors affecting the virulence of *C. neoformans* [69].

## 4. Mitogen-Activated Protein Kinase (MAPK)

The MAPK pathways are critical and evolutionarily conserved mechanisms in eukaryotic organisms, including fungi, animals, and plants. The MAPK component consists of three protein kinases: MAPKKK, MAPKK, and MAPK. Once MAPK is activated, it phosphorylates various substrates and transcription factors. These transcription factors are responsible for rapidly expressing stress-related genes, allowing cells to adapt and thrive under stressful conditions [81]. The MAPK pathways transmit information from extracellular stimuli to the cell, ultimately activating various transcription factors that regulate gene expression in response to these stimuli. MAPK plays a significant role in the physiology and development of fungi, including mating, cell cycle control, morphogenesis, response to different stresses, resistance to UV and temperature variations, cell wall integrity and assembly, degradation of cellular organelles, virulence factors, cell-to-cell signaling, fungus–plant contact, and response to damage (Figure 5) [82,83]. The MAP kinase cascade is the fundamental signal transduction pathway responsible for conserving cell integrity in yeast. In this pathway, the upstream components include membrane sensors that sense stresses to the fungal cell wall [74]. Molecular studies of important medically pathogenic fungi have shown that MAP kinases are also vital virulence factors, because mutant strains defective in them showed reduced virulence in specific animal infection models. Recent studies have revealed that these pathways play an important role in the control of critical virulence factors, for example, capsule production in *C. neoformans* or morphogenesis, spreading, and oxidative stress in *C. albicans* [84].

Recent findings indicate that the MAPK signaling pathway contributes to cell wall integrity and plays a significant role in host infection. The protein Mpk1, part of the cell integrity MAPKK, is essential for cell proliferation at the standard human body temperature of 37 °C. Research on mutant strains lacking Mpk1 suggests that this protein is necessary during host infection and plays a significant role in the pathogen’s virulence. In the fungus *C. albicans*, the gene *MKC1* is vital for cell multiplication at 42 °C but not at 37 °C, indicating an adaptation to its usual host environment compared with the typical saprophytic status of *C. neoformans*. The Mkc1 MAPK is activated in response to various stress factors, including oxidative stress, temperature changes, and cell wall damage, indicating a compromised response to the host’s defenses. While the mutant strain shows reduced virulence in systemic infection, it is not as severe as observed in other mutants related to signal transduction [84,85]. The Pbs2-Hog1 MAPK pathway in *C. neoformans* fungus regulates responses to various stresses, such as UV radiation, osmotic shock, high temperature, and oxidative stress. It also affects sexual growth by inhibiting pheromone production. Deletion mutants of *cpk1* and *ste7* in both mating types show reduced mating capabilities but are not entirely sterile. Additionally, the *cpk1*, *ste7*, and *ste11* mutants are unable to undergo haploid fruiting, but they are as infectious as wild-type strains in animal models such as mice [86]. Understanding the role of the MAPK signaling pathway in *Cryptococcus* pathogenesis is crucial for developing targeted antifungal treatments and vaccines. Current studies aim to uncover the molecular mechanisms underlying MAPK-related virulence in *C. neoformans* and to explore strategies to block these pathways and enhance the body’s defense against fungal diseases.

Current research has expanded our understanding of the intricate roles of MAPK signaling pathways in *C. neoformans*, particularly concerning their involvement in antifungal resistance and intracellular survival. The Cek1 MAPK is known to play a crucial role in regulating antifungal drug susceptibility [87]. The Kpp2 MAPK has been confirmed to support intracellular existence inside macrophages by modifying the expression of genes associated with nutrient acquisition as well as stress response, highlighting its important role in evading host immune response [88]. A recent study has shown that the Hog1 MAPK is activated through environmental stresses and plays an important function in regulating cellular mechanisms, enabling the *C. neoformans* to resist and survive against the host immune defense system [89]. MAPK pathways also play a key role in modulating autophagy, emphasizing the potential of targeting autophagy-related pathways [90]. While many MAPK substrates have been identified, the full range, particularly including those directly linked to virulence factors, still needs to be completed [91]. Despite substantial advancement in understanding the MAPK pathways in *C. neoformans*, there are still quite a lot of crucial features that still need to be explored, underscoring the ongoing nature of this reason. Unexplored research areas in relation to the MAPK pathway include understanding the molecular mechanisms behind its role in fungal virulence, pathogen–host interactions, and developing targeted strategies to block MAPK signaling. The interactions between MAPK pathways and other cellular signaling networks, such as the cAMP/PKA and TOR pathways, remain largely unexplored. Studying these connections could provide insights into how *C. neoformans* integrates diverse environmental signals to coordinate complex cellular responses.

## 5. RAS1 and RAS2 Signal Transduction Pathways

The RAS proteins are small guanine nucleotide-binding proteins that are critical in controlling various cellular functions. In microorganisms, RAS proteins regulate important cell processes such as mating, morphological transitions, metabolism, and pathogenesis [92,93]. These proteins act as molecular switches via their ability to hydrolyze GTP. When bound to GTP, these signaling proteins are active and generate a positive signal by interacting with effectors such as adenylate cyclase [94]. Eukaryotic cells sense and adapt to different environmental changes through conserved signal transduction pathways that link specific extracellular signals to specific cellular responses. These proteins are key regulators of signal transduction pathways that enable adaptive changes, including morphogenesis and cellular development.

It is well established that *C. neoformans* thrives at high temperatures. Ras1 signaling is crucial for thermotolerance, differentiation, and pathogenesis in *C. neoformans* [95]. The Ras1 protein is believed to play a substantial role in these processes, particularly in thermotolerance, as it is a significant virulence factor contributing to pathogenicity [94]. The Ras1 protein also significantly contributes to the activation of both Cdc42 and Rac [95]. In *C. neoformans*, the two crucial Rac paralogues are primarily involved in the polarization and polarized growth of the cell, especially in hyphal development during mating and in the transfer of vesicles during the yeast phase, despite the potential for functional redundancy between Rac and Cdc42. Furthermore, Ras1 signaling through Cdc42 is essential for proper bud shape [58]. The *C. neoformans* Ras2 protein also serves distinct functions. For instance, in a mouse model of cryptococcosis, ∆*rRAS2* cells did not affect pathogenicity [96]. Additionally, compared with any single mutation alone, both *ras1*∆ and *ras2*∆ mutants displayed more significant growth abnormalities at any temperature. These results imply that the functions of Ras1 and Ras2 proteins in mating, growth, and virulence may have some overlapping functions, which necessitates further investigation [97]. The overexpression of the *RAS2* gene completely suppressed the mating failure and partially alleviated the growth defects at high temperatures in the *RAS1* mutant strain. After an extended period at a restrictive temperature, the *RAS1* mutant strain exhibited actin polarity defects, which were partially mitigated by the overexpression of the *RAS2* gene [96]. Even though substantial research has been carried out to understand the significance of Ras1 and Ras2 in the virulence of *C. neoformans*, more than a few features still need more exploration, such as the interaction between RAS pathways and other signaling pathways. For example, the PKC1 pathway that controls cell wall integrity and stress responses still needs to be studied more. Furthermore, the detailed molecular mechanisms by which RAS proteins contribute to immune evasion and biofilm formation within host environments still need to be fully elucidated. Future research should investigate these interactions in detail and assess the possibility of targeting RAS pathways to mitigate fungal virulence.

## 6. Calcium–Calcineurin Signaling Pathway

Calcium (Ca^2+^) is a crucial signaling molecule in living organisms [98]. In the case of *C. neoformans*, several key factors contribute to its ability to establish and cause infection within the host body [7]. These factors are expressed through a complex network of signaling pathways and intracellular processes, in which calcium Ca^2+^ plays a pivotal role. In eukaryotic cells, including fungal cells, calcium signals can regulate various cellular processes, such as responses to temperature, salt, and stress and adaptation to differences in osmotic pressure and pH [99]. Additionally, calcium can influence gene expression, secretion pathways, enzyme activity, protein phosphorylation status, and cell cycle progression [100]. Furthermore, calcium signals are involved in fungal adaptation within the host environment, transmigration mechanisms, survival and growth at 37 °C, virulence, and sexual reproduction [101,102].

In recent years, the calcium signal transduction pathway in fungi has garnered significant attention due to its vital role in fungal survival. It is believed that the filamentation of pathogenic fungi is caused by specific calcium channel proteins, such as calcineurin (CN), which regulates calcium homeostasis in fungi and is known as a virulence factor in fungi [103]. Several research studies have suggested that various components of this critical calcium signaling pathway play key roles in fungal physiology, facilitate stress coping mechanisms, and promote virulence [104,105]. Intracellular Ca^2+^ can bind to polyphosphates, representing the cellular non-reusable level of Ca^2+^ stock. Ca^2+^ homeostasis depends on both active processes and ATP depletion, and its precise regulation is an essential factor for cellular metabolism. Eukaryotic organisms have evolved sophisticated mechanisms to rigorously regulate intracellular calcium levels spatially and temporally, creating an intricate protein network [106]. In *C. neoformans*, the calcineurin signaling pathway is the central responsive pathway, a highly conserved pathway in eukaryotic organisms. This pathway is crucial for yeast cell survival at 37 °C, stress response, sexual reproduction, virulence, and cell wall integrity [107]. The CN signaling pathway is essential for *C. neoformans* pathogenesis and adaptation in various environments [13]. This pathway regulates gene expression, mRNA decay, stability, and various cellular activities. In *C. neoformans*, there are two categories of CBPs. The first category consists of intrinsic transmembrane proteins that transport Ca^2+^ ions through membranes to maintain physiological intracellular Ca^2+^ levels. The second category includes Ca^2+^-modulated proteins, which are typically regulated by sensor proteins and change their conformation upon binding to Ca^2+^ [108]. In *C. neoformans*, the neuronal calcium sensor 1 (Ncs1), a calcium-sensor protein, belongs to the EF-hand superfamily [102].

Similar to its function in *C. neoformans*, CN is vital in virulence and thermotolerance [109]. CN also plays a role in unisexual reproduction. It was determined that this pathway is affected at different phases of sexual reproduction through both Crz1-dependent and Crz1-independent paths [110]. The fungal calcium signaling pathway is crucial for fungal resistance to antifungal medicines [111]. Another new element of the *C. neoformans* Ca^2+^ signaling pathway has been characterized: the vacuolar calcium exchanger (Vcx1). A mutant strain lacking Vcx1 expression had altered CN-dependent Ca^2+^ tolerance and reduced ability to kill mice.

Furthermore, the *VCX1* mutant did not affect cell wall or capsule size but caused reduced secretion of the essential component, polysaccharide glucuronoxylomannan (GXM) [112]. In a 2020 study, it was demonstrated that neuronal calcium sensor 1 (Ncs1) plays a vital role in maintaining calcium balance, cell cycle progression, bud emergence, and virulence. Ncs1 also regulates the CN/Crz1 signaling pathway [102]. High levels of calcium inside fungus cells can be harmful, and several important proteins that control calcium balance, such as channels, transporters, and pumps, are found in the cell membrane of *C. neoformans*. These include Cch1, a calcium voltage-gated channel required for fungus virulence, and a stretch-activated calcium channel [113]. Except for calmodulin, very little is known about the function of other calcium-binding proteins (CBPs) [102]. Pmc1 calcium is crucial for *C. neoformans* migration through the blood–brain barrier (BBB), indicating that Pmc1–calcium balance is essential for the progression of cryptococcosis [114]. CN interacts with various pathways, including MAPK. Disruption of calcium signaling leads to abnormalities in fungal cells, impacting reproductive development, polar growth, differentiation, stress response, and programmed cell death. The interplay between CN and MAPK pathways has been linked to increased stress resilience as well as modified pathogenicity [115]. Recent research highlights that CN plays a role in controlling sexual reproduction via mechanisms that are both dependent on and independent of Crz1 [116]. A supplementary study is required to determine and analyze other calcium-binding proteins plus channels that play a role in the virulence of *C. neoformans*. Hence, the calcineurin signaling pathway is an intriguing target for the development of antifungal drugs.

## 7. Concluding Remarks and Future Directions

The signaling pathways of *C. neoformans* play an important role in its adaptability and virulence, allowing it to survive and thrive in hostile host environments. In this review, we have discussed the protein Kinase A/cAMP, CWI, MAPK, RAS1, RAS2, and calcium–calcineuri pathways. We have thoroughly summarized these pathways, collecting data from the most recent research articles and attempting to explain the links and the significance of each pathway. Each pathway contributes to the regulation of critical virulence factors, including capsule formation, thermotolerance, morphogenesis, and stress response. These pathways either function independently or in coordination, demonstrating the pathogen’s capacity to thrive in hostile host environments, such as the lungs and brain, where it leads to severe and often life-threatening infections. This review highlights the complex nature of these signaling networks and their responsiveness to various environmental signals. Nonetheless, there are still substantial knowledge gaps regarding how these pathways interconnect, which specific signaling cascades prevail during infection, and how *C. neoformans* orchestrates its virulence mechanisms across distinct host tissues. Addressing these unresolved issues requires more focused research to unravel the molecular processes that govern pathway interactions and regulatory dynamics. Future research should focus on several important aspects. One key area involves identifying which signaling pathways are most active during infection, particularly in lung and brain tissues. It is also crucial to understand how various pathways contribute to *C. neoformans* survival, adaptation, replication, and its ability to evade the host’s immune system in different anatomical environments. Another important avenue involves exploration of the mechanisms that allow the pathogen to shift between pathways in response to different stressors. Lastly, clarifying the molecular crosstalk between these signaling networks is essential to uncover how *C. neoformans* fine-tunes its responses for optimal survival and virulence.

## Figures and Tables

**Figure 2 jof-10-00786-f002:**
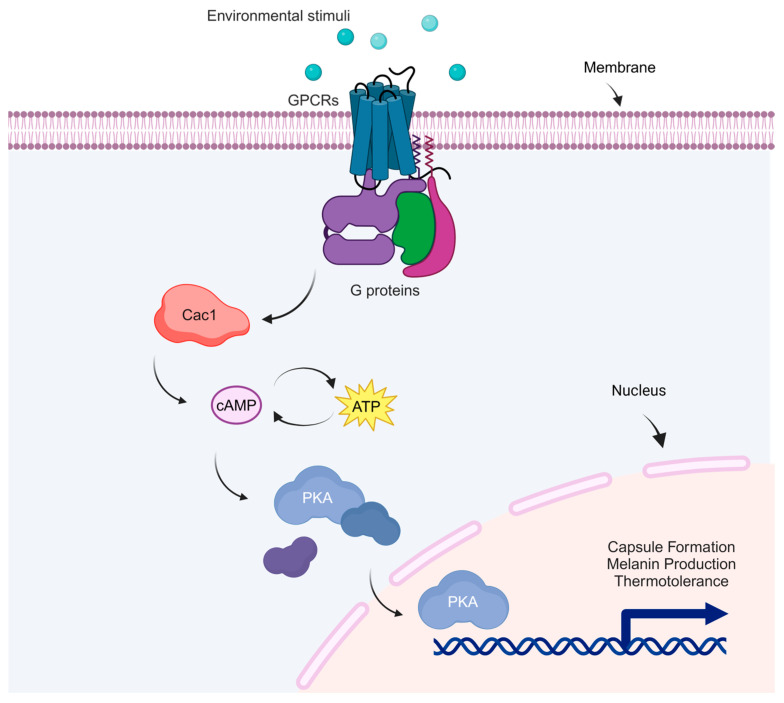
cAMP/PKA signaling pathway in *C. neoformans*. This figure shows the cAMP/PKA pathway in *C. neoformans*. Environmental signals trigger the activation of G-protein-coupled receptors (GPCRs) on the cell membrane. This activation stimulates G proteins, which in turn activate the enzyme adenylyl cyclase (Cac1). Cac1 converts ATP into cAMP, a secondary messenger that activates protein kinase A (PKA). When activated, PKA moves into the nucleus, which controls gene expression, contributing to key virulence factors such as capsule formation, melanin production, and thermotolerance.

**Figure 3 jof-10-00786-f003:**
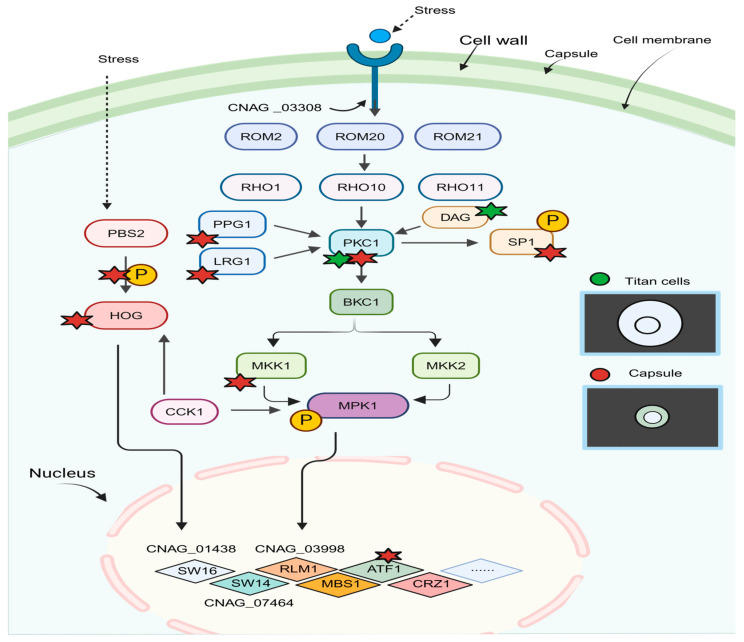
Model of the CWI signaling cascade. The CWI signaling pathway and its impact on *Cryptococcus* morphological changes are currently being studied. In *C. neoformans*, the cell membrane acts as a sensor for environmental stress stimuli, although the specific stimuli are still unknown. It is similar to Mid2. In the *S. cerevisiae* fungus, a transmembrane CWI pathway receptor called CNAG_03308 is present, but its function in the CWI pathway is yet to be discovered. In *C. neoformans*, there are three homologs (Rom20, Rom2, and Rom21) that may activate small GTPases Rho1, Rho10, and Rho11, which then stimulate Pkc1, controlled by Ppg1 and Lgr1. In addition to the GTPases, DAG can also activate Pkc1. Pkc1 plays a crucial role in maintaining the CWI pathway and inducing morphological changes in *C. neoformans*. Several studies have highlighted the central role of Pkc1 in determining cryptococcal morphology, which contributes to cell wall integrity and influences morphological changes in *C. neoformans* to the CWI pathway. In addition, Pkcl phosphorylates the *C. neoformans* transcription factor Sp1, which is important for regulating the expression of *C. neoformans* virulence factors. Downstream of the pathway are transcription factors that maintain the integrity of the cell wall, as shown in the figure. In *C. neoformans*, some components of the CWI pathway are involved in morphological changes in *C. neoformans* cells during interactions with the host: the formation of capsules (red signals) and Titan cells (green signals). Cck1 cytokines also regulate the expression of the HOG pathway and link the CWI pathway to other cell wall integrality-related pathways, contributing to the morphological change of *C. neoformans*.

**Figure 4 jof-10-00786-f004:**
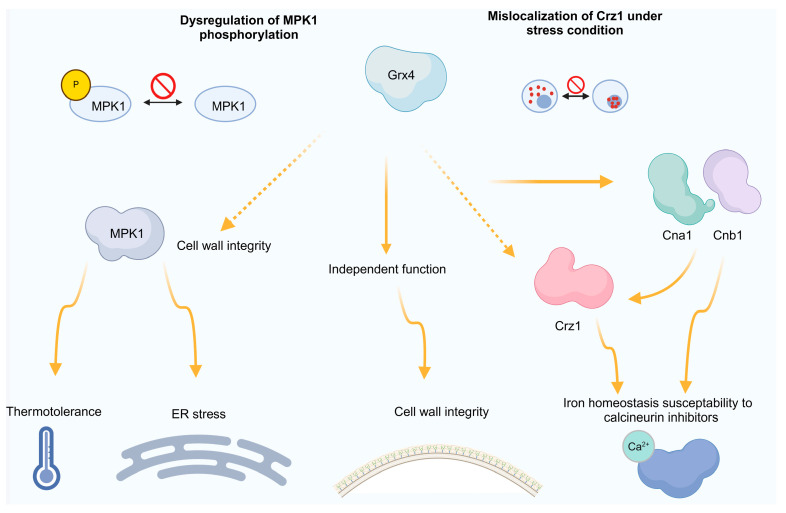
An overview of how Grx4 controls the reactions to stress that are mediated by Mpk1, Crz1, and calcineurin. Grx4 regulates stress responses via Mpk1, with connections to Crz1 and calcineurin. The diagram highlights Grx4’s impact on characteristics related to virulence, such as cell wall structure, thermotolerance, membrane integrity, ER stress, and ion balance. These attributes are associated with the cell wall integrity pathway through Grx4’s influence on Mpk1 activation. Additionally, connections are established between common traits found in mutants with decreased calcineurin signaling and Grx4’s impact on the nuclear translocation of the Crz1 factor [77]. The upstream structure of *C. neoformans* CWI signaling remains unknown [75].

**Figure 5 jof-10-00786-f005:**
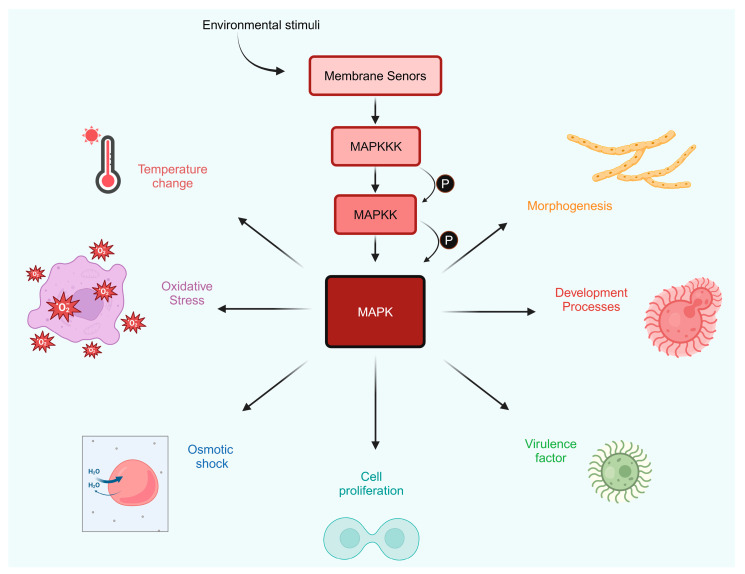
MAPK signaling pathway in *C. neoformans*. This image shows the MAPK pathway, highlighting its significance in *C. neoformans* as a response mechanism to several environmental stimuli. The process starts when membrane sensors detect external signals, such as changes in temperature, osmotic shock, and oxidative stress. These environmental signals start a phosphorylation cascade involving sequential activation of MAPKKK, followed by MAPKK, and culminating in the initiation of MAPK. As soon as MAPK is activated, it orchestrates various cellular responses vital for the survival and pathogenicity of *C. neoformans*. These responses include morphogenesis, virulence factor production, development processes, and cell proliferation, each depicted in the figure with corresponding visual illustrations. Additionally, the figure draws attention to the inhibition of MAPK pathways.

**Table 1 jof-10-00786-t001:** Role cAMP–PKA in different organisms.

S. No.	Organisms	Roles	References
1	Animals	Many cell or tissue-specific functions and processes	[23]
2	*Saccharomyces cerevisiae*	Pseudohyphal differentiation and nutrient sensing	[24]
3	*Schizosaccharomyces pombe*	Germination and mating	[21]
4	*Magnaporthe oryzae*	Formation and function of the appressorium	[25]
5	*Fusarium oxysporum*	Virulence	[26]
6	*Magnaporthe grisea.*	Growth, morphogenesis, and pathogensis	[27,28]
7	*Penicillium oxalicum*	Growth, glucose utilization, and cellulose hydrolysis	[29]
8	*Chaetomium globosum*	Cellulase expression	[30]
10	*Colletotrichum higginsianum*	Pathogenicity, growth, tolerance to cell wall inhibitors, and conidiation	[31]
11	*Fusarium oxysporum*	Penetration into the vascular system and virulence	[32]
12	*Neurospora crassa*	Growth, conidiation, and carbon metabolism	[33]
13	*Ustilago maydis*	Pathogenesis, morphogenesis, and gall formation	[34]
14	*Botrytis cinerea*	Pathogenicity	[34]
15	*Aspergillus fumigatus*	Virulence	[35]
16	*Candida albicans*	Morphogenesis, growth	[36]

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
