# Peer review of "Roles of Different Signaling Pathways in Cryptococcus neoformans Virulence"

_jof, 2024, doi:10.3390/jof10110786_

Round 1

Reviewer 1 Report

In this review submitted by Fawad Mahmood et al., the authors discussed in detail the Different Signaling Pathways in Virulence of Cryptococcus neoformans. C. neoformans is an encapsulated opportunistic microorganism that can be found in environments such as soil, bird droppings, and decaying tree trunks.

This microorganism normally does not cause problems in healthy individuals, but it can become pathogenic in people with compromised immune systems, such as patients with HIV/AIDS or those using immunosuppressants. Infection by Cryptococcus neoformans, called cryptococcosis, primarily affects the central nervous system and can cause meningitis and other serious complications if not treated appropriately. The morphology of this fungus includes yeast-like cells, usually small to medium in size, with a capsule visible on microscopy. Understanding the characteristics, morphology, and pathogenesis of Cryptococcus neoformans is essential for the diagnosis and appropriate treatment of cryptococcosis.

The authors in this review discussed data from more than 124 research articles where it was possible to see that many signaling pathways control various characteristics of C. neoformans, individually or in association with other pathways, and to establish a strong link between all of them to better understand the pathogenesis of C. neoformans.

The review is interesting and very well presented. I only suggest that the authors discuss in more detail the involvement of the capsular polysaccharide, this discussion can happen at the beginning of the review (introduction section).

Authors need to write about the chaosula and polysaccharides and are allowed to add the references they think best.

In this review submitted by Fawad Mahmood et al., the authors discussed in detail the Different Signaling Pathways in Virulence of Cryptococcus neoformans. C. neoformans is an encapsulated opportunistic microorganism that can be found in environments such as soil, bird droppings, and decaying tree trunks.

This microorganism normally does not cause problems in healthy individuals, but it can become pathogenic in people with compromised immune systems, such as patients with HIV/AIDS or those using immunosuppressants. Infection by Cryptococcus neoformans, called cryptococcosis, primarily affects the central nervous system and can cause meningitis and other serious complications if not treated appropriately. The morphology of this fungus includes yeast-like cells, usually small to medium in size, with a capsule visible on microscopy. Understanding the characteristics, morphology, and pathogenesis of Cryptococcus neoformans is essential for the diagnosis and appropriate treatment of cryptococcosis.

The authors in this review discussed data from more than 124 research articles where it was possible to see that many signaling pathways control various characteristics of C. neoformans, individually or in association with other pathways, and to establish a strong link between all of them to better understand the pathogenesis of C. neoformans.

The review is interesting and very well presented. I only suggest that the authors discuss in more detail the involvement of the capsular polysaccharide, this discussion can happen at the beginning of the review (introduction section).

Authors need to write about the chaosula and polysaccharides and are allowed to add the references they think best.

Reviewer 2 Report

This review summarizes a large amount of literature in a field that is worth reviewing, and the rationale for the value of this information is made clear in the abstract and introduction. As such, this paper has a clear purpose and goal. However, it does not achieve that goal due to its confusing organization and writing.

The text is divided into sections for the different signalling pathways, but the text within the sections is disorganized and hard to follow. Some sections are repetitive. The examples below are not a comprehensive list of problems, but a sample.

1) The virulence traits in lines 32-33 are referenced repeatedly later. They should be described in detail in the introduction.

2) Fig 1 caption has confused text in lines 128-129 referring to “a second messenger” that is actually cAMP. Line 132: What are “projectiles”? The arrow between AMP and cAMP is going the wrong way. Pk1 should be Pka1.

3) The lists of “unexplained areas” in Fig. 2 and 5 are useful but should be expanded as text, not as part of a figure. Similar lists should be made for each of the pathway sections.

4) The paragraph in lines 214-229 is very confused and needs re-writing. “Upsurges” (line 215) and “pouring” (line 216) need to be replaced with better words. There are incomplete sentences (line 226) and jumps between unrelated topics (titan cells and iron).

5) The entire section on Cell Wall Integrity (line 243, not “Cell Integrity”) in confusingly written. Lines 264-267 are repeated from earlier in the paper. “Core kinases” in line 284 are not identified. Text between lines 295-301 is repetitive. Lines 314-325 ore confusingly organized, and lines 318-321 are redundant. Lines 335-340 are not related to signalling pathways.

6) Fig 3 contains proteins not explained in the text.

7) The purpose and relevance of Fig. 4 is not made clear, or even which section or pathway it relates to.

8) Lines 370-372 are repetitive.

9) The genes listed in lines 396-397 are not described.

10) Lines 445-449 are out of place.

11) Lines 507-510 appear to have been re-worded from sources about different fungi.

The bibliography is not useful because ALL of the journal names are either missing or incorrectly formatted. The relevance of each reference to its citation should also be checked: Ref. 36 is the wrong reference.

Reviewer 3 Report

 The review broadly summarizes the various signaling pathways described so far for C. neoformans with the aim of understanding its pathogenesis, thereby facilitating the potential development of new antifungals, diagnostic methods, and prevention strategies. This article provides an overview of the molecular processes of C. neoformans, analyzing key signaling pathways such as PKA, PKC, MAPKs, Ras, and the involvement of calcineurin. I believe its contribution is significant to the scientific community

There is an error in line 241. It should be changed to cAMP/PKA

Reviewer 4 Report

General the manuscript is written well , providing informative details about different pathways for virulence and pathogenesis

No major comments detected 

It’s recommended if possible to edit figures to have more details, to make it more understandable and easy to follow 
